# Local E-rhBMP-2/β-TCP Application Rescues Osteocyte Dendritic Integrity and Reduces Microstructural Damage in Alveolar Bone Post-Extraction in MRONJ-like Mouse Model

**DOI:** 10.3390/ijms25126648

**Published:** 2024-06-17

**Authors:** Anh Tuan Dang, Mitsuaki Ono, Ziyi Wang, Ikue Tosa, Emilio Satoshi Hara, Akihiro Mikai, Wakana Kitagawa, Tomoko Yonezawa, Takuo Kuboki, Toshitaka Oohashi

**Affiliations:** 1Department of Molecular Biology and Biochemistry, Okayama University Graduate School of Medicine, Dentistry and Pharmaceutical Sciences, Okayama 700-8558, Japan; tuananh.dang2309@gmail.com (A.T.D.); wangziyi@s.okayama-u.ac.jp (Z.W.); a.mikai@s.okayama-u.ac.jp (A.M.); p7nq13m5@s.okayama-u.ac.jp (W.K.); tomoy@cc.okayama-u.ac.jp (T.Y.); oohashi@cc.okayama-u.ac.jp (T.O.); 2Department of Oral Rehabilitation and Regenerative Medicine, Okayama University Graduate School of Medicine, Dentistry and Pharmaceutical Sciences, Okayama 700-8525, Japan; de421035@s.okayama-u.ac.jp (I.T.); kuboki@md.okayama-u.ac.jp (T.K.); 3Department of Oral Rehabilitation and Implantology, Okayama University Hospital, Okayama 700-8558, Japan; 4Cartilage Biology and Regenerative Medicine Laboratory, Section of Growth and Development, Division of Orthodontics, College of Dental Medicine, Columbia University Irving Medical Center, New York, NY 10032, USA; 5Advanced Research Center for Oral and Craniofacial Sciences, Okayama University Graduate School of Medicine, Dentistry and Pharmaceutical Sciences, Okayama 700-8558, Japan; haraemilio@okayama-u.ac.jp

**Keywords:** medication-related osteonecrosis of the jaw, BMP-2, osteocyte dendritic network, microcrack accumulation, bone remodeling

## Abstract

The pathology of medication-related osteonecrosis of the jaw (MRONJ), often associated with antiresorptive therapy, is still not fully understood. Osteocyte networks are known to play a critical role in maintaining bone homeostasis and repair, but the exact condition of these networks in MRONJ is unknown. On the other hand, the local application of E-coli-derived Recombinant Human Bone Morphogenetic Protein 2/β-Tricalcium phosphate (E-rhBMP-2/β-TCP) has been shown to promote bone regeneration and mitigate osteonecrosis in MRONJ-like mouse models, indicating its potential therapeutic application for the treatment of MRONJ. However, the detailed effect of BMP-2 treatment on restoring bone integrity, including its osteocyte network, in an MRONJ condition remains unclear. Therefore, in the present study, by applying a scanning electron microscope (SEM) analysis and a 3D osteocyte network reconstruction workflow on the alveolar bone surrounding the tooth extraction socket of an MRONJ-like mouse model, we examined the effectiveness of BMP-2/β-TCP therapy on the alleviation of MRONJ-related bone necrosis with a particular focus on the osteocyte network and alveolar bone microstructure (microcrack accumulation). The 3D osteocyte dendritic analysis showed a significant decrease in osteocyte dendritic parameters along with a delay in bone remodeling in the MRONJ group compared to the healthy counterpart. The SEM analysis also revealed a notable increase in the number of microcracks in the alveolar bone surface in the MRONJ group compared to the healthy group. In contrast, all of those parameters were restored in the E-rhBMP-2/β-TCP-treated group to levels that were almost similar to those in the healthy group. In summary, our study reveals that MRONJ induces osteocyte network degradation and microcrack accumulation, while application of E-rhBMP-2/β-TCP can restore a compromised osteocyte network and abrogate microcrack accumulation in MRONJ.

## 1. Introduction

Medication-related osteonecrosis of the jaw (MRONJ) is characterized by exposed and/or non-exposed necrotic alveolar bone in the maxilla or mandible in patients who previously or are currently undergoing antiresorptive therapy for a period longer than 8 weeks without radiation in the head–neck region [1,2]. MRONJ causes significant physical and psychological impairments, leading to negative impacts on patients’ quality of life in addition to social exclusion [3,4,5].

Currently, the pathology of MRONJ is not fully understood. The inhibition of osteoclast activity has been considered to play a central role in the onset of MRONJ [6,7]. In addition, bisphosphonates are known to reduce the activity of osteoblasts, thereby impairing overall bone remodeling [8,9]. Another important cell in bone remodeling is the osteocyte. Osteocytes are embedded in the bone matrix and can communicate with their neighbor cells through a complex network of dendritic processes extending from their cell bodies through the multiple branching canaliculi. It has been demonstrated that this lacunar–canalicular network acts as a mechano-sensing system [10,11,12] that regulates bone remodeling not only by controlling osteoblast and osteoclast function but also by modulating the gradient of ionic substitutions in bone mineralization [13,14,15]. Importantly, previous studies have indicated the association of osteocyte death and empty osteocyte lacunae with impaired bone remodeling and jawbone necrosis in MRONJ [16,17,18]. However, the exact conditions and roles of osteocytes in MRONJ remain unclear. Therefore, a more detailed analysis of the osteocyte network and bone microstructure is crucial to evaluate the conditions of the alveolar bone in the onset of MRONJ.

On the other hand, although various therapeutics for MRONJ have been developed, no gold standard treatment for MRONJ has been established [19,20,21]. Bone morphogenetic proteins (BMPs) constitute the largest group of the transforming growth factor-β (TGF-β) superfamily, which is well known as the multi-functional growth factor that is critically involved in both neonatal and postnatal development. At present, over 20 human BMP subtypes have been identified and characterized since the first investigations that were conducted in the 1960s [22,23,24]. Among those, BMP-2 has been proven to be one of the most promising candidates for bone regeneration [25,26]. The United States Food and Drug Administration (FDA) approved mammalian cell-derived recombinant human BMP-2 (C-rhBMP-2, INFUSE, Medtronic Sofamor Danek, Memphis, TN, USA) for use in spine surgery in mature patients in 2002 [27,28,29]. The application of C-rhBMP-2 in clinical oral and maxillofacial osseous defects has been allowed since 2007 [30,31]. Our research group has also achieved recognition in producing Good Manufacturing Practices (GMP)-grade rhBMP-2 using an Escherichia coli production system (E-rhBMP-2) at the commercial level. The E-rhBMP-2 has been proven to exhibit biological activities similar to those of the C-rhBMP-2 at a lower cost [32,33]. Furthermore, we also demonstrated that E-rhBMP-2 adsorbed onto beta-tricalcium phosphate (E-rhBMP-2/β-TCP) could significantly induce bone formation around dental implants in a swine sinus lift model and a canine-guided bone regeneration model [34,35]. Recently, we reported that the implantation of E-rhBMP-2/β-TCP into the tooth extraction socket not only promoted bone regeneration within the extraction socket but also improved the osteonecrotic condition of the surrounding bone in a MRONJ-like model in mice [36,37]. Nevertheless, the effects of E-rhBMP-2/β-TCP on osteocytes have not been evaluated in MRONJ-like models.

Therefore, in this study, we aimed to clarify the detailed changes that occur in the osteocyte network and bone microstructure of the alveolar bone surrounding the tooth extraction socket in MRONJ. Furthermore, we hypothesized that MRONJ-related decrease in bone remodeling and osteocyte disarrangement could be restored by E-rhBMP-2/β-TCP treatment.

## 2. Results

### 2.1. E-rhBMP-2/β-TCP Treatment Supports the Recovery of the MRONJ-Induced Degeneration of the Osteocyte Network

First, to clarify the three-dimensional (3D) organization of the osteocyte dendritic network in MRONJ, we performed z-stack confocal imaging of the alveolar bone stained with phalloidin. In contrast with the massive osteocyte dendritic network observed in the healthy group, the MRONJ group displayed an extreme reduction in the number and connectivity of this network. Occasionally, cell bodies without dendrite (yellow arrowhead) were found in the MRONJ group (Figure 1). Interestingly, the group treated with E-rhBMP-2/β-TCP showed a complete restoration of the osteocyte dendritic network to levels similar to the healthy group.

Thereafter, using the reconstructed data, we performed a quantitative analysis of the 3D osteocyte dendritic network based on the Filament Tracer function of Imaris (Figure 2A, Appendix A). A statistical analysis confirmed the significant decrease in the osteocyte process volume (µm^3^/cell) and length (µm/cell), the main process number, and the total branch number per individual osteocyte in the MRONJ group compared to the healthy group (*p* < 0.001). Similar trends were found in both the crestal and apical regions. On the other hand, the same parameters were completely restored in the E-rhBMP-2/β-TCP treatment group, resembling those in the healthy group. Indeed, the values for the osteocyte process length and number as well as the total branch number in the group treated with E-rhBMP-2/β-TCP were even significantly higher compared to those in the healthy group (*p* < 0.05) (Figure 2B–E).

### 2.2. E-rhBMP-2/β-TCP Treatment Ameliorates MRONJ-Induced Suppression of Alveolar Bone Remodeling

Previous studies have demonstrated the important roles of osteocytes in modulating bone remodeling [38]. Therefore, to validate whether the application of E-rhBMP-2/β-TCP affects bone remodeling surrounding the tooth extraction socket in an MRONJ-like mouse model, a calcein double bone labeling assay was carried out, as illustrated in Figure 3A. The representative images (Figure 3B) show a clear difference in the intensity of the calcein signal at the tooth extraction socket. As shown in Figure 3C, the two green lines surrounding the tooth extraction socket represent the boundaries of the newly mineralized bone after calcein administration. Notably, these two lines could be observed surrounding the tooth extraction socket of the control and E-rhBMP-2/β-TCP treatment groups. On the other hand, disconnected single lines or no line could be detected surrounding the tooth extraction socket in the MRONJ group, indicating that there was no active bone formation.

To confirm these findings, we further performed measurements for the percentage of mineralized surface per bone surface (MS/BS %) and the mineral apposition rate (MAR, µm/day) using the representative images shown in Figure 4A and Figure 4C, respectively. Statistical analyses (Figure 4B,D,E) confirmed that, at both the apical and crestal regions, there was no bone generated in the MRONJ group (mean bone formation rate (BFR) at apex or crest = 0 µm^3^/µm^2^/day). These results were significantly lower than those of the healthy group (BFR: apex = 0.33 ± 0.08 µm^3^/µm^2^/day, crest = 0.22 ± 0.08 µm^3^/µm^2^/day). In contrast, the E-rhBMP-2/β-TCP treatment group showed a significantly higher (*p* < 0.001) BFR both at the apex (0.51 ± 0.21 µm^3^/µm^2^/day) and crest (0.43 ± 0.12 µm^3^/µm^2^/day) compared to the other counterparts.

### 2.3. E-rhBMP-2/β-TCP Treatment Eliminates MRONJ-Induced Alveolar Bone Micro-Damage Accumulation

Finally, an SEM analysis was carried out to investigate the microstructure of the alveolar bone surrounding the tooth extraction socket at the apical and crestal regions in the healthy, E-rhBMP-2 treatment, and MRONJ groups. Representative pictures were taken at the apical and crestal regions of the alveolar bone surrounding the tooth extraction socket at 60×, 300×, and 3000× magnifications (Figure 5).

At 300× magnification, the tooth extraction socket of the MRONJ group showed abnormal bone healing. The socket was mostly filled with low-contrast soft tissue at both the apical and crestal regions. On the other hand, the healthy and E-rhBMP-2/β-TCP treatment groups showed tooth sockets that were entirely occupied by dense and high-contrast trabecular bone.

At 3000× magnification, the MRONJ group displayed a rough, porous surface with an irregular and disorganized bone architecture surrounding the tooth extraction socket. Notably, there was a large number of microcracks (yellow arrowhead) and bone pealing along the bone surface, which are typically found in necrotic bone [39,40]. In contrast, both the healthy and E-rhBMP-2/β-TCP treatment groups displayed a bone microstructure with a smoother and seamless surface, rarely containing microcracks.

To examine those damages more accurately, an in-depth microcrack analysis was performed using the Filament Tracer function of the Imaris software and SEM images with 3000× magnification (Figure 6A). Notably, the results revealed a similar trend in all microcrack parameters in the MRONJ group, which were significantly higher than those in the healthy and E-rhBMP-2/β-TCP treatment groups (Figure 6B–D). At the apical region, the MRONJ group showed a significant (*p* < 0.001) increase in all microcrack parameters (mean area = 103.7 ± 27.33 µm^2^, mean length = 13.50 ± 2.12 µm, and mean diameter = 0.35 ± 0.05 µm) compared to the healthy group (mean area = 3.19 ± 5.05 µm^2^, mean length = 2.14 ± 1.56 µm, and mean diameter = 0.11 ± 0.09 µm). Nevertheless, there was no significant difference between the E-rhBMP-2/β-TCP treatment group (mean area = 13.77 ± 20.84 µm^2^, mean length = 2.43 ± 3.06 µm, and mean diameter = 0.10 ± 0.12 µm) and the healthy group in all parameters (*p* > 0.05). At the crestal region, the MRONJ group only displayed a significant increase in the microcrack area (µm^2^) compared to the other two groups (*p* < 0.001).

## 3. Discussion

We previously reported that E-rhBMP-2/β-TCP implantation does not enhance the healing of a tooth extraction socket under normal conditions, but it promotes substantial bone regeneration in the extraction socket and improves osteonecrosis of the surrounding bone in MRONJ [36,37]. In the present study, we examined the pathogenesis of MRONJ with a particular focus on osteocyte network integrity and bone microdamage formation. Our findings reveal that MRONJ is characterized by an increased occurrence of microcracks and a compromised osteocyte network. More interestingly, the application of E-rhBMP-2/β-TCP could restore the osteocyte network to a level comparable to that of an extraction socket healed under normal conditions and abrogate microcrack damage induced by MRONJ. These results suggest that E-rhBMP-2/β-TCP may not only play an important role in bone repair but may also be a therapeutic agent capable of repairing MRONJ-induced bone microdamage.

Osteocytes are unique cells whose function critically relies on their communication with other cells via their lacunar–canalicular network. This complex network is believed to be a key player in mechano-transduction, which greatly contributes to the repair of bone microdamage by modulating the bone remodeling process [41,42,43]. Indeed, Kurata et al. and Hazenberg et al., in their study on the relationship between osteocytes, microcracks, and bone metabolism in the MLO-Y4 osteocyte-like cell line, indicated that damage to osteocyte processes can result in the release of RANKL, which stimulates the differentiation of osteoblasts and osteoclasts, promoting microcrack repairment [41,44]. Results from other in vivo models further support this hypothesis [45,46,47]. It is also known that RANKL produced by osteocytes is essential for the maturation of osteoclasts [48]. However, the relevance of osteocyte dendritic connectivity and bone remodeling in MRONJ disease remains unknown. Our 3D osteocyte dendritic analysis revealed a dramatic reduction in the osteocyte process number, volume, and length, as well as in the total branch number (per individual osteocyte) in the MRONJ group compared to that in the healthy controls. Furthermore, significant differences were found in both the apical and crestal regions, indicating that the changes in the osteocyte dendritic network are location-independent. These results are consistent with those of Tiede-Lewis et al. and Kobayashi et al., who used samples from aged mice [49] and patients with osteoporosis [50], respectively. On the other hand, implanting E-rhBMP-2/β-TCP into the extraction sockets restored the osteocyte dendritic network, and this was expected to significantly impact osteoclast behavior. Further investigation is required to elucidate how the administration of E-rhBMP-2/β-TCP to the extraction sockets alters RANKL production by osteocytes and subsequently influences osteoclast activity. This detailed analysis is crucial to understanding the full impact of E-rhBMP-2/β-TCP treatment on MRONJ.

Bone remodeling is a complex biological process that involves the breakdown of old bone by osteoclasts and the formation of new bone by osteoblasts [51,52,53]. In the pathology of MRONJ, bisphosphonates and anti-RANKL antibodies are known to reduce the activity of not only osteoclasts but also osteoblasts both in vitro and in vivo, thereby impairing bone remodeling and increasing the incidence of bone necrosis [9,54]. Our data revealed a delay of bone remodeling in the MRONJ group, which showed significantly decreased MS/BS (%), MAR (µm/day), and BFR (µm^3^/µm^2^/day) values compared to those of the healthy group (*p* < 0.001). This is consistent with the results from the research by Huja et al. [55] and Howie et al. [56], which was conducted using mice and rats, respectively. On the other hand, BMP-2 is recognized for its ability to stimulate osteoblast-mediated bone formation via the canonical and non-canonical signaling pathways of Smad [57,58]. Supporting this, our group has previously reported that the expression of osteopontin—a non-collagenous protein secreted by osteoblasts—in the tooth extraction socket was higher in the E-rhBMP-2/β-TCP transplanted group compared to the MRONJ group [36]. Interestingly, in the present study, the E-rhBMP-2/β-TCP-treated group showed significant improvements in all bone formation parameters compared to the MRONJ group (*p* < 0.001). These findings have strengthened the evidence regarding the osteogenic potential of BMP-2 in inducing osteoblast activity to ameliorate the MRONJ-induced suppression of bone metabolism.

Microcracks are defined as damage in the form of microscopic sharp edges with lengths ranging from 30 to 100 µm, which are larger than osteocyte canaliculi but smaller than transverse vessels [59,60]. Generally, the appearance of microcracks in the alveolar bone is a consequence of the high intensity and frequency of mechanical loading mainly originating from the mastication forces [61,62]. In healthy bone, those microcracks are continuously growing and being repaired by the physiological bone remodeling process [63,64]. Notably, Li et al. and Kim et al. indicated that the microcrack density and length were significantly higher in MRONJ than in the healthy controls [65,66]. Furthermore, Ha et al. showed that the necrotic bone remained on the surfaces of failed implants removed from patients treated with bisphosphonates and antiresorptive agents, and they exhibited hardened bone tissues with microcracked bony resorbed lacunae [67]. Consistently, our present study revealed an abnormal bone structure characterized by a larger microcrack area and longer length in the MRONJ group compared to the healthy group. Notably, the mean microcrack length in our study is shorter than that reported previously [65]. This is in line with a study by Taylor et al., which indicated that a bone from larger animals has higher fatigue strength than that from smaller animals, leading to a higher percentage of microcrack density and an increased accumulation of microcracks [68]. Moreover, bone microstructural studies indicate that microcracks in bones are longer when viewed longitudinally than transversely. This might explain the observation of shorter microcracks in this study compared to other investigations, where the analysis plan was perpendicular to the alveolar Haversian canal [69]. In particular, the group treated with E-rhBMP-2/β-TCP exhibited a restoration in all evaluated microcrack parameters compared to the healthy group. Based on these findings, it could be inferred that a significant reduction in bone metabolism may correlate with an increased incidence of microcracks in the alveolar bone of MRONJ, which can be ameliorated by the administration of E-rhBMP-2/β-TCP.

## 4. Materials and Methods

### 4.1. BMP-2 Materials

The E-rhBMP2/β-TCP combination was prepared by adding 2.5 µL of 0.5 mM HCl containing 2.5 µg of BMP-2 (Osteopharma Inc., Osaka, Japan) to 1.5 mg of β-TCP (Superpore^®^, particle size of 0.6–1.0 mm, porosity of 75%, HOYA, Tokyo, Japan). Before trans-plantation into the first molar tooth extraction socket in the mice, the mixture was incubated for five minutes at room temperature to ensure the E-rhBMP-2 solution was absorbed in the porous β-TCP.

### 4.2. Generating the MRONJ-like Mouse Model

C57BL/6J mice (8-week-old to 12-week-old females) were purchased from CLEA Japan Inc. (Osaka, Japan). The generation of an MRONJ-like model in mice was performed using a combined administration of cyclophosphamide (CY) and zoledronate (ZA), as described [36,37]. Briefly, ZA (0.05 mg/kg, Zometa, Novartis, Stein, Switzerland) and CY (150 mg/kg, C7397; Sigma-Aldrich, St. Louis, MO, USA) were injected, respectively, subcutaneously and intraperitoneally, twice a week for three weeks. Subsequently, maxillary first molars were extracted under general anesthesia with isoflurane (Pfizer, New York, NY, USA) (Figure 7A), and CY and ZA were injected twice a week for an additional two weeks after tooth extraction. After a total of 5 weeks of CY/ZA administration, the tooth extraction sockets were curetted by a dental probe, and E-rhBMP-2/β-TCP was transplanted into the tooth extraction sockets. CY/ZA administration was terminated at the same time as the transplantation of E-rhBMP2/β-TCP. The MRONJ and BMP-2 treatments were performed in the same mouse to avoid inter-individual differences (more details in Figure 7C,D).

The animal experiment protocols (OKU-2020380 and OKU-2022242) were approved by the Okayama University Research Committee. All animals were handled according to the guidelines of the Okayama University Animal Research Committee.

### 4.3. Calcein Double Bone Labeling Experiment

To label active bone formation, a 2 mg/mL calcein solution was prepared by dissolving 0.01 g of sodium bicarbonate and 0.02 g of calcein (C0875-5G, Sigma-Aldrich, Schnelldorf, Germany) in 10 mL of 0.9% sterile saline. The calcein solution was injected intraperitoneally (0.1 mL per 10 g of body weight) in the mice twice, at 10 days and 3 days before euthanasia.

To collect the tooth extraction socket specimens, the mice were deeply anesthetized and then submitted to the intracardiac perfusion of 0.1 M pH 7.4 phosphate-buffered solution (PBS), followed by a solution of 4% paraformaldehyde (PFA) in PBS at 4 °C. After perfusion, the maxilla was collected and further immersed in 4% PFA for at least 2 days at 4 °C and protected from light.

### 4.4. Osteocyte Dendrite Network Staining with Phalloidin

Frozen sections were prepared using Kawamoto’s film method. At first, fixed samples were immersed in 15% and 30% sucrose solutions at 4 °C for 24 h for cryoprotection. To prepare frozen blocks, the samples were embedded in Super Cryo-embedding Medium (SECTION-LAB Co. Ltd., Hiroshima, Japan) and deep-frozen in hexane (Sigma-Aldrich, St. Louis, MO, USA) immersed in dry ice. Frozen sections (7 μm thick) were then made using an adhesive film (Cryofilm type 2C (9) C-396 FP094, SECTION-LAB Co. Ltd., Hiroshima, Japan). Sections were fixed in 4% PFA and washed 3 times in PBS. Fixed sections were decalcified with Osteosoft^®^ (Sigma-Aldrich, St. Louis, MO, USA) for 15 min before being incubated in 0.3% Triton-X100 (Sigma-Aldrich, St. Louis, MO, USA) for 3 minutes. Subsequently, sections were incubated in Histo One Blocking solution (06349-64, Nacalai, CA, USA) for 10 minutes and then incubated with Alexa Fluor 546-conjugated Phalloidin (1:500; A22283, Invitrogen, Waltham, MA, USA) for 2 h at 37 °C. After incubation, the sections were washed in PBS, counter-stained with DAPI (Vector Laboratories, Newark, CA, USA), and mounted using Fluoromount-G^®^ (Southern Biotech, Birmingham, AL, USA). The stained sections were used for both the bone formation analysis and confocal microscope imaging as described below.

### 4.5. Bone Formation Analysis

Images were taken with a fluorescence microscope (Keyence BZ-X710, Osaka, Japan). Mineralized surface per bone surface (MS/BS, %) was measured by dividing the calcein-positive area per total bone area (calibrated with the FOV of the image). The calcein-positive area was defined using the color threshold (RGB) function of Image J software (version 1.53k, NIH, Maryland, USA). Mineral apposition rate (MAR, μm/day) was calculated by dividing the average distance between the two calcein labeling lines by the interval time of administration (7 days). Bone formation rate (BFR, μm^3^/μm^2^/day) was calculated as MAR*(MS/BS) according to a previous study [70].

### 4.6. Confocal Microscope Imaging

Two comparable regions of interest (ROIs) at the crestal and apical regions of the alveolar bone surrounding the tooth extraction socket were selected for analysis. The calcein labeling green lines used to evaluate the dynamic bone formation were defined based on the boundaries of the tooth extraction sockets.

Z-stack images were captured using a confocal laser scanning microscope system (Zeiss LSM 780, Carl Zeiss, Germany) with a 63×/1.4na oil immersion objective lens (Plan-Apochromat DIC M27, W Korr, Carl Zeiss, Germany) with the following optical parameters: 3-channel acquisition mode, namely DAPI (410–476 nm)/calcein (489–561 nm)/Phalloidin Alexa Fluor 546 (568–735 nm). The confocal microscope was available at the Central Research Laboratory, Okayama University Medical School. Detector gains (master) and digital offsets were adjusted to maximize the dynamic range within a given image stack, and a linear gain/offset compensation was employed to ensure a consistent image signal. Photobleaching was minimized by protecting the sections from light in all staining steps as well as by controlling the excitation intensity level of the laser while performing the confocal microscope capture.

### 4.7. Scanning Electron Microscopy (SEM)

The coronal section of the maxilla was prepared by cutting the specimen through the maxillary first molars position using a low-speed handpiece with a diamond disc (H306F220, Horico, Germany) at a speed of 2000 rotations per minute (rpm) with water irrigation to avoid overheating. The sample was then fixed in 4% PFA in PBS at 4 °C for at least 24 h. To standardize the surface roughness, the samples were wet-polished using an MA-200D polishing machine (Musashino Denshi, Tokyo, Japan) with a diamond polishing magnetic plate (Grid size: 35 µm and 15 µm, 41-2739-208-001, Apex B, Buehler, Lake Bluff, IL, USA) at a speed of 120 rpm for 30 s per sample. Each sample was then kept in a separated poly-ethylene zip-lock bag, followed by immersion in distilled water in an ultrasonic cleaner (UP-7 SPC seria, Shinka, Japan) at sound pulses of 40 kHz at room temperature for 15 min to remove the debris remaining on the bone surface following the previous cleaning protocol [71].

For SEM observation, samples were dehydrated with a serial grade of ethanol for 30 min for each step and then replaced with 100% ethanol and t-butanol. The samples were then placed onto an aluminum holder and submitted to 2% Osmium (Os) coating (Neoc-STB, Meiwafosis, Tokyo, Japan) before observation using an ultra-high resolution field emission scanning electron microscope (SEM) (Model S-4800 HITACHI, Tokyo, Japan).

### 4.8. Image Processing Using Zen, ImageJ, and Imaris Software

For analysis of the osteocyte dendritic network, z-stack confocal fluorescence images were first processed in the Zen 2012 SP1 software (black edition version 8.1, Carl Zeiss, Germany) to produce the maximal projection. The files were then exported in tiff format using ImageJ software. Subsequently, the images were imported into Imaris (version 9.3.0, Oxford Instruments, UK) for the 3D reconstruction of osteocyte morphology and analysis. Algorithms were created with the Filament Tracer function to render and measure the 3D osteocyte dendritic processes’ network parameters, which included the process volume (µm^3^/cell), process length (µm/cell), main process number, and total branch number. The main process number was counted using a custom R programming language script based on the exported Imaris results (available at https://github.com/LabOnoM/IMAO (accessed on 9 April 2024)).

A microcrack analysis was initially performed using the Image J software to make the inverted layer of the original 3000× magnification SEM images. Microcracks were determined as the sharp edge and over-exposed linear shape area (white area). Both layers were then overlapped and used to measure the microcrack area (µm^2^), length (µm), and diameter (µm) using a similar Filament Tracer function in Imaris.

### 4.9. Statistical Analysis

Statistical analyses were conducted using Prism 9 (GraphPad Software Inc., La Jolla, CA, USA). Statistical significance between the groups was assessed by using a one-way ANOVA with Tukey’s test. A *p*-value of <0.05 was considered significant. Statistical significances were described as follows: * *p* < 0.05, ** *p* < 0.01, and *** *p* < 0.001. The results were reported as the mean values ± standard deviation (SD). All data points represent biological replicates.

## 5. Conclusions

In conclusion, our study uncovered a novel pathological characteristic in the alveolar bone affected by MRONJ, where the osteocyte network is disrupted, bone remodeling is delayed, and microcracks are considerably accumulated. Importantly, we demonstrated that the transplantation of E-rhBMP-2/β-TCP not only promotes bone regeneration but can also effectively reverse these pathological changes in osteocytes and the bone microstructure. While our data present promising results, further examinations are necessary to gain a more comprehensive understanding of the functional mechanisms of BMP-2 therapy in MRONJ.

## Figures and Tables

**Figure 1 ijms-25-06648-f001:**
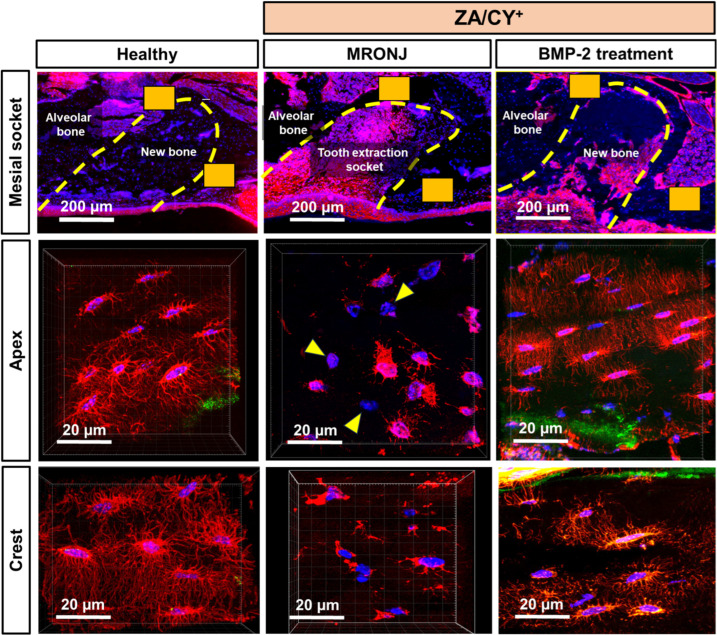
Confocal microscope images of phalloidin-stained sections. Cryosections (7 µm thick) were stained with phalloidin Alexa Fluor 546 (red) to visualize the actin filaments and were counter-stained with DAPI (blue) to visualize the nuclei. The bone (the border of the tooth extraction socket) was indicated with calcein (green). The images in the upper panel illustrate 2 regions of interest (ROI, solid orange boxes), which were the apical and crestal regions within the tooth extraction socket corresponding to the mesial root of the maxillary first molar. Occasionally, cell bodies without dendrite (yellow arrowhead) were seen in the MRONJ group. The maximal z-projection illustrates the changes in the morphology of the osteocyte dendritic network.

**Figure 2 ijms-25-06648-f002:**
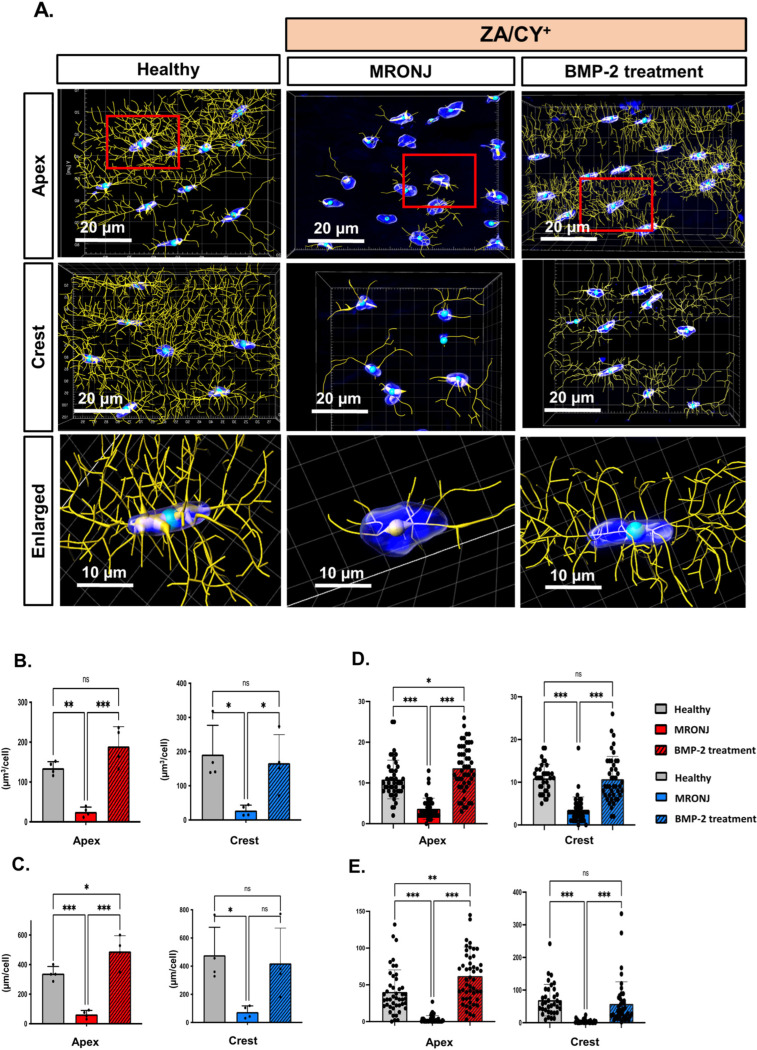
A three-dimensional (3D) reconstruction analysis of the osteocyte network. (**A**) A 3D reconstruction of the osteocyte network using the Imaris software. The enlarged images show a 3D rendering of a single osteocyte, confirming the massive reduction in osteocyte dendritic connectivity in the MRONJ group compared to the healthy and BMP-2 treatment groups. The red boxes indicate the area selected from the tile images. The Filament Tracer function was used to determine the osteocyte processes (yellow) and DAPI-stained nuclei (blue). (**B**) Process volume (µm^3^/cell). (**C**) Process length (µm/cell). (**D**) Main process number (per cell). (**E**) Total branch number (per cell). The data are presented as the mean ± standard deviation (SD); ns: no significant difference. * *p* < 0.05; ** *p* < 0.01; *** *p* < 0.001. One-way ANOVA, Tukey’s test.

**Figure 3 ijms-25-06648-f003:**
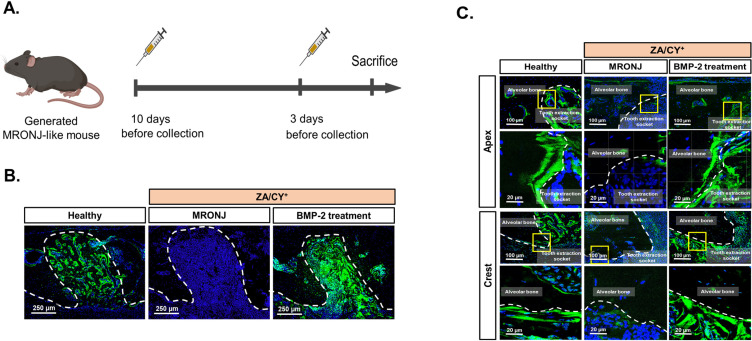
Calcein double bone labeling for the assessment of new bone formation. (**A**) A schematic depicting the experimental design for calcein administration on an MRONJ-like mouse model. Calcein was administered 10 and 3 days before the tissues were harvested. (**B**) The localization of the calcein signal at the tooth extraction socket corresponding to the mesial root of the first molar. (**C**) Representative images demonstrating the double labeling lines (green) that can be observed surrounding the tooth extraction socket of the healthy and BMP-2 treatment groups. The MRONJ group showed a very weak signal and disconnected labeling lines. The dashed lines (white) represent the border between the alveolar bone and the tooth extraction socket. The yellow boxes indicate the selected areas shown in the high-magnification images (lower panels). Calcein (green) and DAPI (blue) indicate the bone and nuclei, respectively.

**Figure 4 ijms-25-06648-f004:**
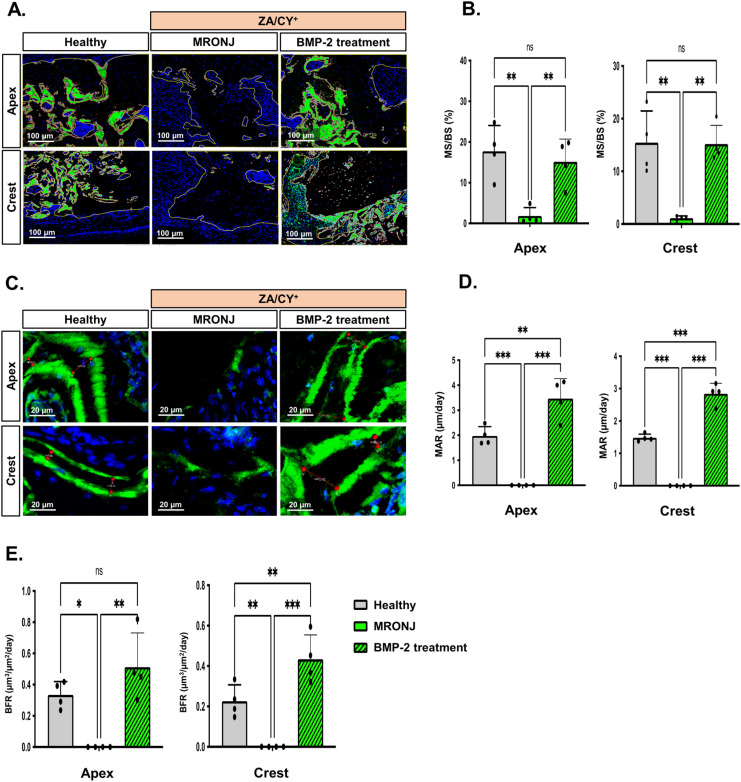
Calcein double bone labeling parameters. (**A**) The method for measuring the percentage of mineralized surface per bone surface (MS/BS %). The yellow lines indicate the total bone area (µm^2^), while the red lines indicate the area (µm^2^) of the newly formed bone. (**B**) The graphs show the results of MS/BS (%) at the apical and crestal regions. (**C**) The method for measuring the mineral apposition rate (MAR) (µm/day). (**D**) The graphs show the results of the MAR (µm/day) at the apical and crestal regions. Since the green line could not be observed in the MRONJ group, the MAR value in MRONJ was determined as 0 μm/day. (**E**) The graphs show the results of the bone formation rate (BFR) (µm^3^/µm^2^/day). The data represent the mean ± standard deviation (SD); ns: no significant difference. * *p* < 0.05; ** *p* < 0.01; *** *p* < 0.001. One-way ANOVA; Tukey’s test. In (**A**,**C**), calcein (green) and DAPI (blue) indicate the bone and nuclei, respectively.

**Figure 5 ijms-25-06648-f005:**
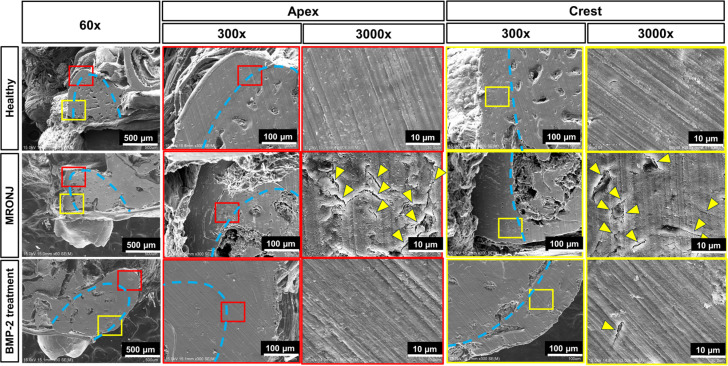
Images taken by a scanning electron microscope (SEM) showing microcrack accumulation in the alveolar bone surrounding the tooth extraction socket. Low-magnification (60×) images showing the region of interest (red and yellow squares) for the microcrack analysis at the apical and crestal regions of the alveolar bone surrounding the tooth extraction sockets. Images at magnifications of 300× and 3000× showing a rough and porous bone surface with an irregular and disorganized bone architecture. The microcracks (yellow arrowheads) and pealing on the bone surface were highly found in the MRONJ group. The dashed lines in blue indicate the border of the tooth extraction socket. The red and yellow boxes indicate the images at the apical and crestal regions, respectively.

**Figure 6 ijms-25-06648-f006:**
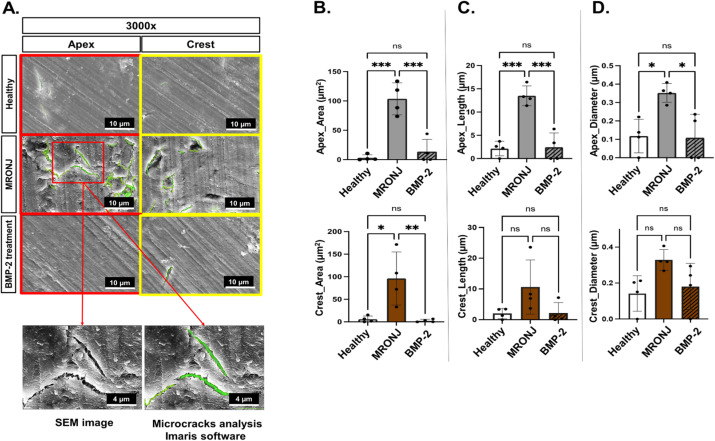
Microcrack analysis. (**A**) SEM images (at 3000× magnification) depicting microcracks. Filament Tracer function of Imaris software was used to determine microcrack (**B**) area (µm^2^), (**C**) length (µm), and (**D**) diameter (µm) at apical and crestal regions. Data represent mean ± standard deviation (SD); ns: no significant difference. * *p* < 0.05; ** *p* < 0.01; *** *p* < 0.001. One-way ANOVA; Tukey’s test.

**Figure 7 ijms-25-06648-f007:**
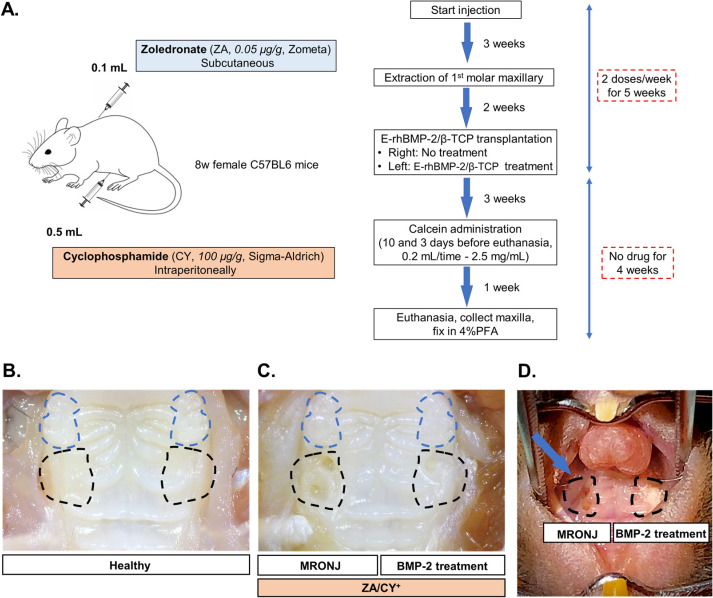
Experimental design: An MRONJ-like model was generated and BMP-2 treatment was used in mice. (**A**) A combination of cyclophosphamide (CY) and zoledronate (ZA) was administered twice weekly for three weeks to generate an MRONJ-like model in mice. At the end of this period, the maxillary first molars were extracted, and CY/ZA administration was continued for another 2 weeks. After 5 weeks of CY/ZA administration, the tooth extraction sockets were curetted, and E-rhBMP-2/β-TCP was transplanted into the tooth extraction sockets. CY/ZA administration was terminated at the time of E-rhBMP2/β-TCP implantation. Tissues were harvested 4 weeks after E-rhBMP-2/β-TCP transplantation. Calcein was administered 10 and 3 days before the tissues were harvested. Intraoral pictures of the (**B**) healthy (positive control) and (**C**) MRONJ groups without (left) or with (right) E-rhBMP2/β-TCP treatment. (**D**) An image depicting the site where E-rhBMP-2/β-TCP was transplanted at the tooth extraction socket in the mice. In (**B**,**C**), the dashed lines in blue indicate the 2nd and 3rd molars. In (**B**–**D**), the dashed lines in black indicate the tooth extraction sockets in the area corresponding to the first molar.

## Data Availability

Data are contained within the article.

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
