# Peer review of "Local E-rhBMP-2/β-TCP Application Rescues Osteocyte Dendritic Integrity and Reduces Microstructural Damage in Alveolar Bone Post-Extraction in MRONJ-like Mouse Model"

_ijms, 2024, doi:10.3390/ijms25126648_

Round 1

Reviewer 1 Report

Comments and Suggestions for Authors

In this article, the authors developed a model of MRONJ in female mice treated with anticancer and antiresorptive drugs, followed by tooth extraction. The animals were then treated with E-rhBMP-2/β-TCP to improve healing. I have some comments and questions regarding the methodology:

Methods: In the calcein images, are the cell nuclei stained with DAPI? This detail is not specified in the methodology section.

Section 4.4: The term "Bone mineralized" is used. Did the authors intend to refer to "parameters of bone mineralization" or "bone mineralization analysis"?

It appears that one socket received the E-rhBMP-2/β-TCP treatment while the other did not in both the Healthy and MRONJ groups. However, only the treated sockets for the MRONJ groups are presented. It is crucial to include a control group that also received treatment for a comprehensive comparison. Please present this data, as the absence of controls significantly limits my ability to accurately analyze the results and also the authors ability to support their conclusions.

Comments on the Quality of English Language

Must be revised.

Author Response

Photos of previous data have been uploaded as pdf files.

Could you check PDF file?  Best regards.

Reviewer #1

Comment #1

Section 4.4: The term "Bone mineralized" is used. Did the authors intend to refer to "parameters of bone mineralization" or "bone mineralization analysis"?

Response #1

Thank you for your comment. We changed it to “bone formation analysis”.

Comment #2

It appears that one socket received the E-rhBMP-2/β-TCP treatment while the other did not in both the Healthy and MRONJ groups. However, only the treated sockets for the MRONJ groups are presented. It is crucial to include a control group that also received treatment for a comprehensive comparison. Please present this data, as the absence of controls significantly limits my ability to accurately analyze the results and also the author's ability to support their conclusions.

Response #2

Thank you for your precise comments. The Reviewer is correct. There is no E-rhBMP-2/β-TCP administered to the healthy group in this study. However, we have previously reported two papers to clarify the effect of E-rhBMP-2/β-TCP on MRONJ (Mikai et al., IJMS, 2020, Tanaka et al., IJMS, 2021). In the first paper, we confirmed that E-rhBMP-2/β-TCP administered into the tooth extraction socket of a healthy group did not enhance bone formation in the tooth extraction socket (Mikai et al, IJMS, 2020, right Figure). Therefore, in the subsequent studies in this series, the analysis has been performed without the healthy group treated with E-rhBMP-2/β-TCP. Since this was not fully explained in this paper, we have now described in the main text why these three groups were used in this study (Discussion section, first paragraph, lines 246-257).

  1. The first paragraph of the discussion was changed to explain that implantation of BMP-2 in the normal extraction socket does not significantly affect wound healing in the normal extraction socket.

“We have previously reported that E-rhBMP-2/β-TCP implantation does not enhance the healing of a tooth extraction socket under normal conditions, but promotes substantial bone regeneration in the extraction socket and improves osteonecrosis of the surrounding bone in MRONJ [36][37]. In the present study, we examined the pathogenesis of MRONJ with a particular focus on osteocyte network integrity and bone microdamage formation. Our findings revealed that MRONJ is characterized by an increased occurrence of microcracks and a compromised osteocyte network. More interestingly, the application of E-rhBMP-2/β-TCP could restore the osteocyte network to a level comparable to that of an extraction socket healed under normal conditions, and abrogate microcrack damage induced by MRONJ. These results suggest that E-rhBMP-2/β-TCP may not only play an important role in bone repair but may also be a therapeutic agent capable of repairing MRONJ-induced bone microdamage.”

  1. At the end of the session on "Generating the MRONJ-like mouse model" in methods, the following is described.

 “The MRONJ and BMP-2 treatment groups were performed in the same mouse to avoid inter-individual differences (more details in Figure 1C, D).”

  1. Figures 1b and c were modified to more clearly show the experimental design. As shown below, the positive control is the healthy group, the negative control is MRONJ, and the experimental group is the BMP-2 treatment group, as shown in Figure Legend 1.

“Intraoral pictures of the (B) healthy (positive control) and (C) MRONJ groups without (left) or with (right) E-rhBMP2/β-TCP treatment. (D) Image depicting the site where E-rhBMP-2/β-TCP was transplanted at the tooth extraction socket in mice. For (B-C), dashed lines in blue indicate the 2nd and 3rd molars. For (B-D), dashed lines in black indicate the tooth extraction sockets in the area corresponding to the first molar.”

Reviewer 2 Report

Comments and Suggestions for Authors

The topic is of great interest, the study is well conducted and presented. Please find attached the pdf with some comments. 

In particular, I would suggest the Authors to discuss the interrelationship among osteocytes and osteoclasts, explaining why authors did not include the measurement of viable osteoclasts in their analysis.

Author Response

Thank you so much for your reviewing. Below is the point-by-point response to the reviewer’s comments.

We are looking forward to receiving a positive reply at your earliest convenience.

Reviewer #2

Comment #1

In particular, I would suggest the Authors discuss the interrelationship between osteocytes and osteoclasts, explaining why the authors did not include the measurement of viable osteoclasts in their analysis.

Response #1

Thank you for your very important comment. In the second paragraph of the discussion, we have amended the description about the relationship between osteocytes and osteoclasts. However, we are currently preparing to conduct this analysis in more detail and have not been able to clarify this issue in this study, therefore we are aiming to clarify it in future works. Therefore, the following sentences were added in the discussion section (Line 266-281).

“It is also known that RANKL produced by osteocytes is essential for the maturation of osteoclasts [71]. However, the relevance of osteocyte dendritic connectivity and bone remodeling in MRONJ disease has remained unknown. Our 3D osteocyte dendritic analysis revealed a dramatic reduction in the osteocyte process number, volume, and length, as well as total branch number (per individual osteocyte) in the MRONJ group compared to the healthy controls. Furthermore, significant differences were found in both the apical and crestal regions, indicating that the changes in the osteocyte dendritic network are location-independent. These results are consistent with those of Tiede-Lewis et al. and Kobayashi et al. using samples from aged mice [50] and osteoporosis patients [51], respectively. On the other hand, implanting E-rhBMP-2/β-TCP into the extraction sockets restored the osteocyte dendritic network and would be expected to significantly impact osteoclast behavior. Further investigation is required to elucidate how the administration of E-rhBMP-2/β-TCP to the extraction sockets alters RANKL production by osteocytes and subsequently influences osteoclast activity. This detailed analysis is crucial to understanding the full impact of E-rhBMP-2/β-TCP treatment on MRONJ.”

Comment #2

Methods: In the calcein images, are the cell nuclei stained with DAPI? This detail is not specified in the methodology section.

Response #2

Thank you for the detailed comment. In this study, the calcein images and the phalloidin-stained osteocyte images were taken using the same sections, but using two different microscopes for the two independent analyses. The methods were not accurately described, thus, we have now rearranged the order and described them correctly. Moreover, the following text has been added at the end of subsection 4.4 to indicate that the same sample was used for the analysis.

" The stained sections were used for both the bone formation analysis and confocal microscope imaging as described below.”

Additionally, in the legend for Figures 4 and 5, it is noted that blue corresponds to nuclear staining with DAPI:

“Calcein (green) and DAPI (blue) indicate bone and nuclei, respectively.”

Comment #3

Regarding comments in the uploaded pdf

Response #3

Thank you for reviewing the manuscript in detail. We have checked the comments in the PDF and corrected them accordingly.

Line 20: bisphosphonate therapy → antiresorptive therapy

Line46-47: a non-viable exposed maxillary or mandibular alveolar bone → the exposed and/or non-exposed necrotic alveolar bone in the maxilla or mandible

Line 49: without radiation treatment → without radiation in the head-neck region

Line 279: bisphosphonates are known to reduce the activity of osteoblasts both in vitro and in vivo, thereby impairing bone remodeling and increasing the incidence of bone necrosis  → bisphosphonates and anti-RANKL antibodies are known to reduce the activity of not only osteoclasts but also osteoblasts both in vitro and in vivo, thereby impairing bone remodeling and increasing the incidence of bone necrosis

Line 282: reveals → revealed

Round 2

Reviewer 1 Report

Comments and Suggestions for Authors

The authors had successfully addressed my comments